# Gut Microbiota and Plasma Bile Acids Associated with Non-Alcoholic Fatty Liver Disease Resolution in Bariatric Surgery Patients

**DOI:** 10.3390/nu15143187

**Published:** 2023-07-18

**Authors:** Álvaro Pérez-Rubio, Polina Soluyanova, Erika Moro, Guillermo Quintás, Iván Rienda, María Dolores Periañez, Andrés Painel, José Vizuete, Judith Pérez-Rojas, José V. Castell, Ramón Trullenque-Juan, Eugenia Pareja, Ramiro Jover

**Affiliations:** 1Servicio de Cirugía General y Aparato Digestivo, Hospital Universitario Dr. Peset, 46017 Valencia, Spain; alvaroprubio@gmail.com (Á.P.-R.); lopego30@hotmail.com (M.D.P.); ramontrullenque@gmail.com (R.T.-J.); eugepareja@outlook.es (E.P.); 2Experimental Hepatology Joint Unit, Health Research Institute La Fe-University of Valencia, 46026 Valencia, Spain; polina_soluyanova@externos.iislafe.es (P.S.); erika_moro@externos.iislafe.es (E.M.); jose.castell@uv.es (J.V.C.); 3Departamento de Bioquímica y Biología Molecular, Universitat de València, 46010 Valencia, Spain; 4Health and Biomedicine, Leitat Technological Center, 08225 Terrassa, Spain; gquintas@leitat.org; 5Pathology Department, Hospital Universitario y Politécnico La Fe, 46026 Valencia, Spain; ivanrienda@gmail.com (I.R.); judithp_r@hotmail.com (J.P.-R.); 6Section of Abdominal Imaging, Radiology Department, Hospital Universitario Dr. Peset, 46017 Valencia, Spain; 7CIBERehd, Instituto de Salud Carlos III, 28029 Madrid, Spain

**Keywords:** NAFLD resolution, gut microbiota, bile acid, bariatric surgery, morbid obesity, Bacteroides, Akkermansia

## Abstract

Bariatric surgery (BS) has several benefits, including resolution of non-alcoholic fatty liver disease (NAFLD) in many patients. However, a significant percentage of patients do not experience improvement in fatty liver after BS, and more than 10% develop new or worsening NAFLD features. Therefore, a question that remains unanswered is why some patients experience resolved NAFLD after BS and others do not. In this study, we investigated the fecal microbiota and plasma bile acids associated with NAFLD resolution in twelve morbidly obese patients undergoing BS, of whom six resolved their steatosis one year after surgery and another six did not. Results indicate that the hallmark of the gut microbiota in responder patients is a greater abundance of Bacteroides, Akkermansia, and several species of the Clostridia class (genera: Blautia, Faecalibacterium, Roseburia, Butyricicoccusa, and Clostridium), along with a decreased abundance of Actinomycetes/Bifidobacterium and Faecalicatena. NAFLD resolution was also associated with a sustained increase in primary bile acids (particularly non-conjugated), which likely results from a reduction in bacterial gut species capable of generating secondary bile acids. We conclude that there are specific changes in gut microbiota and plasma bile acids that could contribute to resolving NAFLD in BS patients. The knowledge acquired can help to design interventions with prebiotics and/or probiotics to promote a gut microbiome that favors NAFLD resolution.

## 1. Introduction

Liver disease has been reported in up to 80% of obese patients who meet the criteria for bariatric surgery (BS), including 65.9% with steatosis and 14.3% with non-alcoholic steatohepatitis (NASH) and/or fibrosis [1].

BS induces long-term excess weight loss, diabetes mellitus remission, and reduced cardiovascular and cancer-related mortality [2]. Moreover, several prospective and retrospective cohort studies and meta-analyses have demonstrated that sustained weight loss after BS is also associated with non-alcoholic fatty liver disease (NAFLD) improvement, including reduced steatosis, inflammation, and fibrosis [2].

In a systematic review and meta-analysis to evaluate the effects of BS on NAFLD in obese patients (including data from 32 cohort studies comprising 3093 biopsy specimens), BS resulted in the resolution of steatosis in 66% of patients [3]. Moreover, it was shown that both of the most common forms of BS (Roux-en-Y gastric bypass and sleeve gastrectomy) are of comparable high efficacies in improving NAFLD [4]. However, steatosis did not resolve in a significant fraction of patients and, in 12%, BS resulted in new or worsening features of NAFLD [3].

One question that remains unanswered is why, after BS, some patients experience resolved baseline NAFLD, while others do not.

It has been reported that persistent moderate or severe steatosis after BS is more frequently observed in patients who have maintained a higher insulin resistance index (1/QUICKI) than in patients who have improved their insulin resistance index (44% vs. 20.2%; *p* = 0.04) [5]. However, this study focused on patients with severe steatosis, which do not represent most BS patients with NAFLD. A preoperative hypocaloric diet is currently recommended for patients undergoing BS to reduce the liver volume and facilitate surgery. This diet usually leads to a reduction in the hepatic steatosis score [6]; therefore, patients with severe steatosis are rare at the time of surgery.

The mechanisms leading to the improvement and resolution of NAFLD after BS are not fully understood, but they may be related to anatomical and endocrine-metabolic changes, as well as to a decrease in portal influx of free fatty acids and a reshaping in gut microbiota and circulating bile acids (BA) [7].

The gut microbiota plays a key role in the homeostasis of the gut-liver axis, and several studies have shown that gut microbial dysbiosis may be involved in the pathogenesis of NAFLD, through different mechanisms such as increased energy harvesting and altered microbial products [8]. Conversely, specific changes in the gut microbiota and microbial products are associated with metabolic improvement in NAFLD patients undergoing BS [9].

Therefore, it is likely that gut microbiota remodeling in patients who show complete resolution of steatosis after BS (responders) is different from that in patients who show persistent steatosis (non-responders). To test this possibility, we investigated the fecal microbiota and plasma BA profiles of two small groups of BS patients (responders and non-responders), at the time of surgery and one year later. Results confirm that responder BS patients show specific changes in fecal microbiota and BA which are not observed in non-responders, suggesting that gut microbiota and BA could be important drivers of NAFLD resolution after BS. Our results could also help to design interventions with prebiotics and/or probiotics to promote a gut microbiome that favors NAFLD resolution.

## 2. Materials and Methods

### 2.1. Patients

This study is a pilot observational study of a larger ongoing project, which included 12 morbidly obese patients recruited between 2020 and 2022 and referred to the Esophageal-Gastric Surgery Section of the Dr. Peset University Hospital in Valencia. Ten patients were Caucasian and two were Amerindian. All the patients fulfilled criteria for BS (Roux-en-Y gastric bypass) according to the 1991 NIH guideline for eligibility [10]. Exclusion criteria included obesity secondary to endocrinopathies, active neoplastic disease, severe psychiatric disorder, alcoholism or drug dependence, non-adherence to recommended hygienic-dietary changes, anesthetic contraindication, and severe comorbidities.

On the day of surgery, a scheduled liver biopsy was taken. Hematoxylin-eosin and Masson’s trichrome-stained paraffin-embedded liver biopsy sections were examined and interpreted by the same experienced hepatopathologists (I.R & J.P.-R.), who were unaware of the patients’ clinical data. Steatosis, along with ballooning, lobular inflammation, and fibrosis, were assessed as outlined by Brunt et al. [11]. Disease severity was scored according to SAF [12]. The % of liver steatosis was also assessed by magnetic resonance imaging (MRI)-proton density fat fraction (PDFF) (mDixonQuant, Philips). The % of liver fat by MRI-PDFF allowed us to divide patients retrospectively into two groups: responders (steatosis was resolved) and non-responders (steatosis was not resolved).

Patient characteristics at the time of bariatric surgery and one year later are presented in Table 1. Informed written consent was obtained from all patients, and the study was conducted in conformity with the Helsinki Declaration and was approved by the Ethics Committee of Dr. Peset University Hospital in Valencia (CEIm n° 44/19).

### 2.2. Metagenomics

Stool samples from patients were collected in Stool Nucleic Acid Collection and Preservation Tubes (Norgen Biotek, Thorold, OT, Canada, Cat. 45660) and kept at 4 °C for up to one month prior to processing.

DNA was extracted using the EZNA^®^ Stool DNA Kit (Omega Bio-Tek, Norcross, GA, USA, Cat. D4015-01) following the manufacturer’s protocol. The concentration and purity of the extracted DNA were determined on a NanoDrop 2000 Spectrophotometer (Thermo Fisher Scientific, Madrid, Spain, Cat. ND-2000).

Metagenomic analyses were performed by SeqPlexing S.L. (Science Park, University of Valencia, Valencia, Spain). The DNA samples were used for 16S amplification of the V3–V4 hypervariable regions. Specifically, metagenomic DNA was amplified using the 341F/805R primer set. Sequencing was performed with the MiSeq Reagent Kit V2 on an Illumina MiSeq System. All samples were multiplexed and sequenced in a single lane on the MiSeq using 2 × 500 bp paired-end sequencing, to a minimum depth of 150,000 reads per sample.

Quality control of sequencing data was performed with FastQC, and MultiQC was used to summarize the analysis results in a single report. Then, the sequencing analysis was performed with the USEARCH v11.13 software, which analyzes the sequences of the 16S gene employing various heuristic computational algorithms in order to determine the zero-radius operational taxonomic units (ZOTUs) present in the samples, the estimate of their abundance, and their taxonomic annotations.

Taxonomy assignation was carried out against HUMAN GUT 16S10, with a 99% alignment cut point (identity) using the algorithm USEARCH-LOCAL1 and then using SINTAX12 (97% identity).

The rarefaction curves were obtained via the core diversity metrics of QIIME 2 to exhibit the sequencing depth of samples. Alpha diversity was calculated using common indices such as the Shannon’s Diversity Index.

### 2.3. Plasma Bile Acid Profiling

Venous blood was collected in glass tubes containing EDTA. Plasma was obtained by centrifugation at 1500× *g* for 10 min at 4 °C and immediately frozen at −80 °C. The concentration of BA in human plasma was determined by UPLC-MS/MS in the Analytical Unit, Core Facility, IIS Hospital La Fe, Valencia. The method [13,14], validated according to FDA guidelines, allows the quantification of 12 non-conjugated, 8 glycine-conjugated, and 11 taurine-conjugated BA, using 5 additional deuterated BA as internal standards in a single analytical run.

### 2.4. Statistics

Continuous normally distributed variables were represented as mean ± standard deviation (SD) (in Table 1) or as mean ± standard error of the mean in figures. Kolmogorov-Smirnov and Shapiro-Wilk tests were used to test the normal distribution of continuous variables. To compare the means of normally distributed variables between groups, the Student’s *t*-test (paired or unpaired, as indicated) was performed. To determine differences between groups for continuous non-normally distributed variables, medians were compared using the Mann-Whitney U test.

## 3. Results

### 3.1. NAFLD Resolution vs. Non-Resolution One Year after BS

Our first objective was to divide BS patients in two groups: responders (R) that resolve NAFLD, and non-responders (NR) that do not resolve NAFLD. To allocate patients in these two groups, we used the % of liver fat determined by MRI-PDFF at time 0 and one year after BS. For this pilot study, we recruited a small group of 12 patients, 6 of them responders and the other 6 non-responders.

At the time of surgery, the two groups of patients showed no significant differences in anthropometric and biochemical parameters (Table 1, t = 0, *p* R vs. NR). Some parameters were slightly higher in R0 (e.g., Triglycerides) whereas others were higher in NR0 (e.g., LDL or insulin), but differences were not statistically significant.

**Table 1 nutrients-15-03187-t001:** Patient characteristics at the time of bariatric surgery and one year later.

	Bariatric Surgery (t = 0)					One Year after Surgery (t = 1)						
	Responders (R)			Non-Responders (NR)		Responders (R)				Non-Responders (NR)		
	Mean	SD	*p* R vs. NR	Mean	SD	Mean	SD	*p* 0–1	*p* R vs. NR	Mean	SD	*p* 0–1
Age (years)	53.7	5.9	NS	51.2	8.2							
Gender female (%)	66.7			66.7								
Heigh (m)	1.64	0.06	NS	1.63	0.12							
Weight max (Kg)	123.3	14.9	NS	121.0	23.1							
BMI max (kg/m^2^)	45.9	4.2	NS	45.6	5.7							
Weight (Kg)	103.1	12.4	NS	102.5	18.3	70.1	14.2	***	NS	83.8	17.6	**
BMI (kg/m^2^)	38.3	2.5	NS	38.8	5.4	26.0	4.1	***	NS	31.8	6.5	**
Glucose (mg/dL)	99.2	15.9	NS	95.0	23.4	97.2	20.8		NS	89.5	15.1	
Urea (mg/dL)	41.2	14.7	NS	38.7	10.9	31.0	13.6		NS	36.5	11.3	
Uric acid (mg/dL)	6.6	2.0	NS	5.2	1.2	4.7	0.8	*	NS	4.7	1.3	*
Cholesterol (mg/dL)	164.4	65.7	NS	180.8	58.7	149.6	20.5		NS	170.2	35.5	
HDL (mg/dL)	36.4	7.2	NS	39.8	12.7	49.8	8.1	*	NS	51.7	14.0	**
LDL (mg/dL)	86.4	66.1	NS	114.7	45.2	82.2	21.1		NS	100.5	28.5	
Triglycerides (mg/dL)	185.8	70.0	NS	130.7	79.5	87.8	20.7	*	NS	88.8	45.3	
AST (IU/L)	16.8	5.4	NS	17.8	7.5	22.6	4.0		NS	20.3	6.7	
ALT (IU/L)	19.8	8.6	NS	24.0	15.7	29.2	13.7		NS	23.7	17.1	
GGT (IU/L)	23.0	3.7	NS	22.7	9.6	24.6	16.9		NS	15.0	6.6	*
ALP (IU/L)	76.6	24.6	NS	84.2	17.7	94.8	21.5		NS	88.3	18.4	
Albumin (g/dL)	4.4	0.4	NS	4.2	0.4	4.3	0.3		NS	4.2	0.4	
Fe (µg/dL)	92.0	35.2	NS	92.0	38.2	92.2	58.9		NS	113.2	21.7	
HbA1c (%)	6.2	1.0	NS	5.8	0.7	5.3	0.5		NS	5.3	0.4	
Insulin (µU/mL)	12.7	5.2	NS	17.9	6.6	5.8	2.4	*	NS	10.6	5.1	*
HOMA-IR	3.5	1.2	NS	4.4	2.2	1.6	1.1	*	NS	2.1	1.0	*
INR	1.0	0.1	NS	1.0	0.1	1.0	0.0		NS	1.0	0.0	
Fibrinogen (g/L)	426.8	66.7	NS	538.0	133.3	484.0	31.1		NS	470.7	66.4	
Hemoglobin (g/dL)	14.7	0.7	NS	13.6	1.1	14.0	1.3		NS	13.9	1.1	
Platelets (10^3^/µL)	268.0	68.1	NS	260.2	47.8	216.8	47.2	*	NS	243.0	32.2	
MRI-PDFF (%)	12.9	9.2	NS	7.7	5.8	1.3	1.0	*	**	5.7	2.2	
Histopathology												
Steatosis (S0/S1/S2/S3)	0/5/1/0			0/5/0/1								
Activity (A0/A1/A2/A3/A4)	1/3/1/0/1			1/3/1/1/0								

* *p* < 0.05; ** *p* < 0.01; *** *p* < 0.001; NS: no significance.

Analyses of liver biopsies demonstrated that all patients had NAFLD, but most of them had mild steatosis (S1) and low activity (ballooning + lobular inflammation) (A1), though few of them showed more severe activity (A2–A4) in both groups (Table 1). In BS patients, the severity of NAFLD is usually reduced by the preoperative hypocaloric diet, in which patients reduce their BMI by 15% (ca. 20 kg).

One year after surgery, the two groups of patients had significantly reduced weight, BMI, insulin, HOMA-IR, triglycerides, and uric acid, and increased HDL cholesterol (Table 1, t = 1, *p* 0–1, paired *t* test). A comparison between responder (R) and non-responder (NR) patients showed that, one year after surgery, the only significant difference was the % of liver steatosis measured by MRI-PDFF (Table 1, t = 1, *p* R vs. NR).

Results in Figure 1 demonstrate that only responder patients significantly reduced their steatosis (from 12.9% at time 0 to 1.3% one year later) (Figure 1A and Table 1), leading to a significant difference of 4.4-fold between R1 and NR1 (Figure 1A and Table 1). On the contrary, BMI was significantly reduced in both groups of patients, and no significant difference between R1 and NR1 was observed one year after BS (Figure 1B and Table 1).

According to previous studies measuring steatosis by MRI-PDFF, the threshold that separates grade 0 steatosis from grades 1–3 is 3.7% [15]. In general, steatosis percentages calculated by MRI-PDFF are lower than histology-calculated percentages (e.g., S0–1 vs. S2–3, histology > 33%, MRI-PDFF > 11%; S0–2 vs. S3 histology > 66%, MRI-PDFF > 20%) [16]. In agreement with this cut-off value, the responder BS group completely resolved their NAFLD (<3.7%) whereas the non-responders slightly improved or did not improve steatosis (>3.7%).

### 3.2. Microbiota Diversity and Number of Bacteria

Microbiomes of fecal samples from BS patients (R and NR) were analyzed by 16S metagenomics, both at time 0 and one year after surgery.

There seems to be a consensus that a decreased gut microbiome diversity is linked to a declined health status. Therefore, alpha diversity, the microbial diversity of an ecological community, was investigated. Regarding community richness (S), which estimates the total number of species (the number of different operational taxonomic units, OTUs, per sample), no significant differences were observed, yet at time 0, community richness was 20% higher in responders than in non-responders (Figure 2A, left). Regarding the Shannon index, which estimates species diversity considering the number of species and their relative abundance, no significant differences were observed either, although again, at time 0, diversity was slightly higher in responders than in non-responders (Figure 2A, right).

The total number of bacteria showed a significant increase in responders from time 0 to one year (Figure 2B). However, no significant differences were observed between responders and non-responders (R0 vs. NR0, or R1 vs. NR1).

### 3.3. Taxonomic Composition

Next, the abundance of bacteria in the different taxonomic levels was investigated. At the phylum level, fecal microbiotas were dominated by two phyla: Bacteroidota (Bacteroidetes) and Bacillota (Firmicutes). Proteobacteria, Actinomycetota, and Verrumicrobiota phyla were also present but with lower abundance (Figure 3A).

An increase of Bacteroidota from time 0 to one year was evident in responders (R1 vs. R0, Figure 3A). A more detailed, paired analysis showed a statistically significant increase (Figure 3B). This change in Bacteroidota was not observed in non-responders (NR) and, consequently, the abundance of Bacteroidota in R1 was significantly higher than in NR1.

An increase in Proteobacteria (3-fold) and a decrease in Actinomycetota (0.5-fold) were also specifically observed in R1 vs. R0 (Figure 3A & Table 2). The decrease of Actinomycetota in responders also led to lower abundance in R1 vs. NR1 (Table 2). Moreover, responders slightly increased the level of Verrucomicrobiota, whereas non responders showed a reduction in this phylum (Figure 3A). Consequently, Verrucomicrobiota was more abundant in R1 than in NR1 (Figure 3A & Table 2).

The increases observed in the phyla Bacteroidota and Verrumicrobiota were maintained in lower taxonomic levels (i.e., Bacteroidia, Bacteroidales, Bacteroidaceae, Bacteroides; and Verrucomicrobiae, Verrucomicrobiales, Akkermansiaceae, Akkermansia). The decreases in the phylum Actinomycetota were also observed in lower taxonomic levels (i.e., Actinomycetes, Bifidobacteriales, Bifidobacteriaceae, Bifidobacterium) (Table 2).

We did not observe significant differences in the phylum Bacillota (Firmicutes). However, we found a substantial increase in R1 vs. R0 in the class Bacilli, order Lactobacillales, and family Streptococcaceae; as well as a substantial decrease in NR1 vs NR0 in the class Negativicutes, order Vellionellales, and family Veillonellaceae; all of them belonging to the Bacillota (Firmicutes) phylum (Table 2).

Three families of the order Bacteroidales showed important increases in R1 vs. R0: Bacteroidaceae, Prevotellaceae, and Odoribacteraceae (Table 2). However, only Bacteroidaceae reached an important higher level in R1 vs. NR1. The family Odoribacteraceae showed increases in both R and NR, one year after BS (Table 2). No differences were observed between responders and non-responders, thus pointing to Odoribacteraceae as an example of gut bacteria induced by other BS-associated factors such as weight loss [17].

Two families from the phylum Bacillota (Firmicutes), Veillonellaceae (class Negativicutes) and Oscillospiraceae (class Clostridia), showed a similar behavior; they were substantially reduced in non-responders one year after BS (NR1 vs. NR0, Table 2).

### 3.4. Analyses of Bacterial Genera and Species

Most genera showing alterations belonged to the classes Bacteroidia (Bacteroides, Prevotella, Butyricimonas, and Parabacteroides) and Clostridia (Roseburia, Anaerobutyricum, Oscillibacter, Faecalicatena, and Butyricicoccus) (Table 2).

Again, several genera (*n* = 7) showed significant differences between R1 and R0, whereas differences between NR1 and NR0 were less frequent (*n* = 3) (Table 2).

Comparative analyses between responders and non-responders showed that, at the time of surgery (t = 0), there were only two genera with greater abundance in R0 vs. NR0: Parabacteroides and Bilophila (Table 2). Bilophila also maintained a greater abundance in R1 vs. NR1 (3.5-fold), although without statistical significance (Figure 4A).

On the contrary, one year after surgery, six genera showed differences (Table 2). Faecalicatena and Bifidobacterium showed decreased levels, whereas Butyricicoccus, Bacteroides, Roseburia and Akkermansia were more abundant in R1 than in NR1 (Table 2 and Figure 4B).

Regarding species, 220 were identified in the metagenomic analyses, but only 138 had reads in ≥50% of patient samples. Among them, 15 species showed significant increases in R0 vs NR0 (at the time of BS) (Table 3, left), including 9 of the genera Bacteroides. Three of these Bacteroides species also showed higher abundance one year after surgery (Table 3, right). Regarding differences between R1 and NR1, 13 species were found with significant increases in R1 (Table 3, right), including 8 from Bacteroides, Blautia, and Clostridium genera.

Some of the most interesting examples of species showing differences between R and NR are *Bacteroides thetaiotaomicron* and *Bilophila wadsworthia*, at time 0 (Figure 4A), and *Blautia obeum* and *Clostridium fessum*, one year after BS (Figure 5B).

Analyses at the species level suggest that the class Bacteroidia, order Bacteroidales, includes most species showing differences between R and NR at time 0 (genera Bacteroides and Parabacteroides), while the class Clostridia, order Eubacteriales, comprises most species showing differences one year after BS (genera Blautia, Faecalibacterium, Roseburia, Butyricicoccus, and Clostridium).

### 3.5. Plasma BA in Responders and Non-Responders

BA were profiled by UPLC-MS/MS in plasma of patients at 0, 3, 6, and 12 months after surgery. Time 0 plasma was collected on the day of intervention, just before surgery.

The results in Figure 6 demonstrate that all patients had an increase in the concentration of plasma BA during the first 6 months after BS, with primary and conjugated BA being responsible for most of this increase. However, from 6 to 12 months after BS, non-responder patients reduced their plasma BA concentrations, whereas responders maintained higher levels. This led to a significantly higher plasma concentration of primary BA in responders 12 months after BS (2.4-fold) (Figure 6, Primary BA). In addition, non-conjugated BA increased steadily in responders, reaching, at month 12, a level 3.6 times higher than in non-responders (Figure 6, Non-conj. BA). The difference is even larger if only primary non-conjugated BA are taken into consideration. In this case, the level in R, at month 12, was 8.2-fold higher than in NR (*p* = 0.099).

A more detailed analysis of BA species at month 12 demonstrated that all primary BA (both non-conjugated and conjugated) had a significantly higher plasma concentration in responders. The highest differences were observed in non-conjugated cholic acid (CA) and chenodeoxycholic acid (CDCA), which were 7- and 11-fold higher, respectively, in responders (Figure 7, left). The levels of secondary BA were much more similar in both groups (Figure 7, right).

## 4. Discussion

BS causes evident changes in the gut microbiome, which could promote some of the BS benefits such as weight loss and NAFLD resolution [7]. We did not observe statistically significant differences in alpha diversity between responders (R, NAFLD resolution) and non-responders (NR). However, at time 0, the Shannon index and species richness were slightly lower in non-responders (NR0), which agrees with other studies showing that patients with persistent NAFLD have lower richness [18].

One of the most remarkable findings of this study was the consistent increase in Bacteroidota (Bacteroidetes) in responders from R0 to R1 and between R1 and NR1 at all taxonomic levels. This was not observed in patients who did not resolve fatty liver (NR). The potentially important role of Bacteroidetes in counteracting NAFLD has also been established in other studies. Del Chierico et al. [19] showed that Bacteroidetes were increased in control healthy patients vs. NAFLD and obese patients, and Mouzaki et al. [20] demonstrated an inverse association between the presence of NASH and the percentage of Bacteroidetes in stools. Similarly, improved intrahepatic triglyceride content based on MRI-PDFF was associated with an increase in the abundance of Bacteroidetes [21]. Thus, a decrease in Bacteroidetes may indicate an increased risk of NAFLD, and probiotics based on Bacteroides could improve NAFLD, as shown in animal models [22]. Bacteroides and other gut bacteria produce short-chain fatty acids (SCFA), which play important roles in gut integrity, lipid metabolism, glucose homeostasis, appetite regulation, and immune responses. However, excessive SCFA may also have harmful effects as they may inhibit AMPK in the liver and increase the accumulation of hepatic free fatty acids, while inducing pro-inflammatory T cells [9].

Another important phylum that showed a significant increase in R1 vs. NR1 at all taxonomic levels was Verrumicrobiota. These changes were attributed to the genus Akkermansia. We and others have already demonstrated a negative association between Akkermansia and hepatic triglycerides in NAFLD, and its potential to ameliorate steatosis [14,23]. Akkermansia is a mucin degrader that has been shown to improve the gut epithelial barrier, reduce organ adiposity, and protect against insulin resistance and obesity in humans [24].

The decrease in the phylum Actinomycetota in patients that resolve fatty liver (R1) (also observed at lower taxonomic levels up to the genus Bifidobacterium) is in agreement with the finding that Actinobacteria is significantly increased in NAFLD and obese patients compared with controls [19]. Regarding Bifidobacterium, it has been associated with probiotics and NAFLD improvement. However, other studies have shown that BS triggers a decrease in Bifidobacterium in stools [25].

Although most changes were observed in responder patients (R1), some interesting alterations were also found in non-responders (NR1). Oscillospiraceae, for example, decreased in non-responders at one year after BS (NR1). This could contribute to fatty liver in non-responders, as a low abundance of the genus Oscillospira has been reported in a group of persistent NAFLD patients [18].

Differences between responders and non-responders are likely the most informative for NAFLD resolution. At the time of surgery (t = 0), we found only two genera with greater abundance in R0 than in NR0: Parabacteroides and Bilophila. The genus Parabacteroides was found to be more abundant in healthy patients than in patients with NAFLD and obesity [19]. Regarding Bilophila, its species *Bilophila wadsworthia* is a very important sulfidogenic bacteria in the human gut. However, it has not been associated with NAFLD improvement [26], yet it has been shown that *Bilophila wadsworthia* infection in mice resulted in a reduction in body weight and fat mass [27].

Comparison of the discriminant bacterial species between R and NR revealed that Clostridia is the class of bacteria with more species showing differences one year after BS, including Blautia, Faecalibacterium, Roseburia, and Butyricicoccus. It has been shown that the most abundant gut genera in the Firmicutes phylum and Clostridia class, Blautia, Faecalibacterium, and Roseburia, are greatly reduced in abundance in obese and NASH patients [28]. Regarding Butyricicoccus, it is a SCFA butyrate-producing genus that can prevent NAFLD progression by maintaining low-grade intestinal inflammation [29]. In addition, it is important to note the decrease in the genus Faecalicatena (Class Clostridia) in R1 vs. NR1, as certain Faecalicatena species have been positively linked to the fatty liver index [30].

Several Bacteroides species showed increased abundance in responders, both at time 0 and one year after BS (*B. faecis*, *B. thetaiotaomicron,* and *B. zhangwenhongii*). *B. thetaiotaomicron* is a dominant bacterial species residing in the human gut (6% of all intestinal bacteria and 12% of all Bacteroides in humans), that exerts important functions in gut mucosal barrier, immunity, and nutrient metabolism. It protected mice against diet-induced obesity [31] and lowered alcohol-induced hepatic steatosis and triglyceride content in *B. thetaiotaomicron*-treated mice [32].

The results indicate that the hallmark of the microbiota in BS patients that resolve fatty liver is a greater abundance of Bacteroides and Akkermansia, as well as several species of the Clostridia class (genera: Blautia, Faecalibacterium, Roseburia, Butyricicoccus, and Clostridium). Moreover, decreased abundance of Actinomycetes/Bifidobacterium and the Clostridia genus Faecalicatena could also contribute to NAFLD resolution.

BA stand out among the most important bacterial products. While the liver is responsible for the production of primary BA, microbes in the gut modify these compounds into many secondary forms that greatly increase BA diversity and biological function. This process is dependent on the metabolic capacity of the bacterial community. The resulting pool of primary and secondary BA have different properties on health outcomes, as they can influence cell signaling, host metabolism, and microbiome composition [33].

We observed a steady increase in total plasma BA levels at 3 and 6 months after BS in both responders and non-responders. Our results are in consonance with previous observations in BS patients and animal models [34,35,36]. Our results also demonstrate that the increase in plasma BA is mostly due to primary conjugated BA.

It can be speculated that the gut microbiota established after BS has less overall BA deconjugating activity. Deconjugated primary BA are transformed into secondary BA by the colon microbiota, whereas conjugated BA are promptly reabsorbed. Thus, the observed increase in plasma BA levels following BS may reflect increased absorption and recirculation through the enterohepatic system. This would also lead to a reduction in total fecal BA, which has also been reported in BS patients and animal models [24,37].

When we compared plasma BA levels in responder and non-responder patients, we only observed differences at 12 months after BS. Responders maintained a high concentration of plasma BA, but non-responders decreased to levels below those at 3 months after BS. One possible explanation is again that responders maintain low BA deconjugating activity. Conjugated BA can be deconjugated by bile salt hydrolase (BSH) activity in bacteria belonging to the genera Lactobacillus, Bifidobacterium, and Enterococcus [38]. In this regard, we found that Bifidobacterium decreased in both R1 vs R0 and R1 vs NR1, which points to a potential source of lower BSH activity in responders, which completely resolved fatty liver.

Deconjugated BA can undergo a variety of microbiota-mediated transformations (i.e., 7 α-dehydroxylation, dehydrogenation, and epimerization) that generate secondary BA, which have widespread effects on the gut microbiota. A large fraction of secondary BA is lost in feces, whereas the opposite is true for primary BA [39]. Thus, an increase in primary plasma BA levels could also be due to a reduced synthesis of secondary BA in the gut. Secondary DCA and LCA are produced from free (deconjugated) CA and CDCA by 7α/β-dehydroxylation activities encoded in BA-inducible (bai) genes. Several Clostridium species, including *C. scindens*, *C. hiranonis*, and *C. hylemonae,* contain such bai genes. However, some Faecalicatena species, which are more prevalent and abundant in humans, harbor putative bai operons to convert primary to secondary BA [40]. Interestingly, we found that the genus Faecalicatena was reduced in R1 vs. NR1, which would imply lower bai activities for secondary BA synthesis in responder patients, after 12 months of BS. Other bacterial species may also contribute to the higher levels of circulating primary BA. We demonstrated that administration of *Akkermansia muciniphila* to rats after HFD was associated with NAFLD amelioration, along with an increase in primary non-conjugated BA such as CA and CDCA [14].

However, the association between BA and NAFLD remains controversial. Several studies have reported that NAFLD is associated with an increase in total BA, but others have concluded that there are no differences between NAFLD patients and controls [41]. Therefore, our observation that increased plasma BA is associated with NAFLD resolution does not agree with some studies showing the opposite: that NAFLD patients show increased plasma BA. However, important differences could exist in the molecular mechanisms underlying plasma BA induction, and in the specific BA species increased. In studies reporting that NAFLD is associated with more BA, the increase observed is usually attributed to primary and secondary conjugated BA [41]. However, we found that most of the increase in BA in responders could be only attributed to primary BA (secondary BA did not increase). Moreover, maximal differences were observed in non-conjugated CA and CDCA. It is also worth noting that CDCA, CA, and their conjugates are strong FXR activators [42] and that activation of hepatic FXR represses lipogenic genes, whereas activation of intestinal FXR reduces lipid absorption [43].

Thus, our results on BA suggest that NAFLD resolution is associated with a sustained increase in primary (particularly non-conjugated) BA, which likely results from a reduction in bacterial gut species capable of generating secondary BA (e.g., Bifidobacterium and Faecalicatena). Signaling of primary BA, such as CDCA, through nuclear receptors, such as FXR, could contribute to a more efficient NAFLD resolution. FXR regulates multiple lipid pathways that can reduce hepatic steatosis. FXR inhibits fatty acid and triglyceride biosynthesis. CDCA lowers the expression of endogenous SREBP-1c and its lipid synthesis target genes in primary hepatocytes. In vivo, the expression of endogenous SREBP-1c was negatively correlated with the amount of CDCA. Incubation of HepG2 cells and human primary hepatocytes with CDCA resulted in upregulation of PPARα and its target gene CPT1 (rate-limiting for mitochondria fatty acid oxidation), whereas in FXR-deficient ob/ob mice, the expression of PPARα was downregulated [44].

From this pilot study we conclude that some morbidly obese patients do not resolve fatty liver after BS. Comparative analyses between responder and non-responder patients have demonstrated specific changes in gut microbiota and plasma BA. From these preliminary observations, it is tempting to suggest that interventions aimed at enriching the gut microbiota with bacteria associated with NAFLD resolution (e.g., Bacteroides or Akkermansia), as well as with FXR agonists, could help to achieve maximum rates of NAFLD resolution in BS patients.

## Figures and Tables

**Figure 1 nutrients-15-03187-f001:**
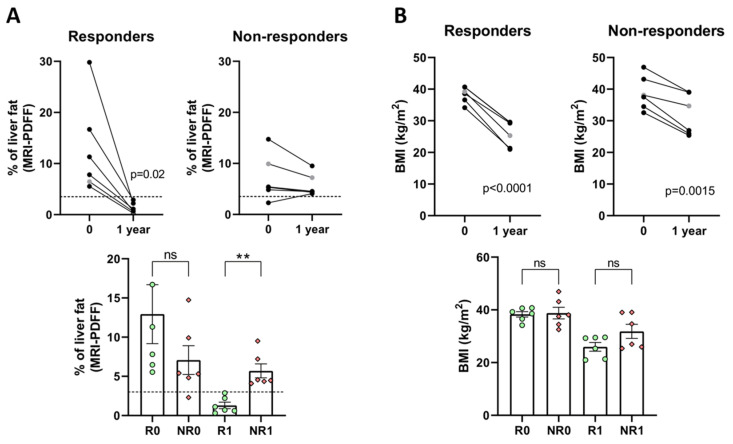
Changes in % of liver fat (**A**) and in BMI (**B**) in responder (R) and non-responder (NR) BS patients. Upper panels: black dots, Caucasian; grey dots, Amerindians. Lower panels: green dots, responders; red diamonds, non-responders; dotted line in (**A**), MRI-PDFF cut-off value for no steatosis (S0). Analyses were performed at the time of surgery (R0 and NR0) or one year later (R1 and NR1). Paired (upper panels) or unpaired (lower panels) *t*-tests were applied to determine statistical significance; ** *p* < 0.01, ns: no significance.

**Figure 2 nutrients-15-03187-f002:**
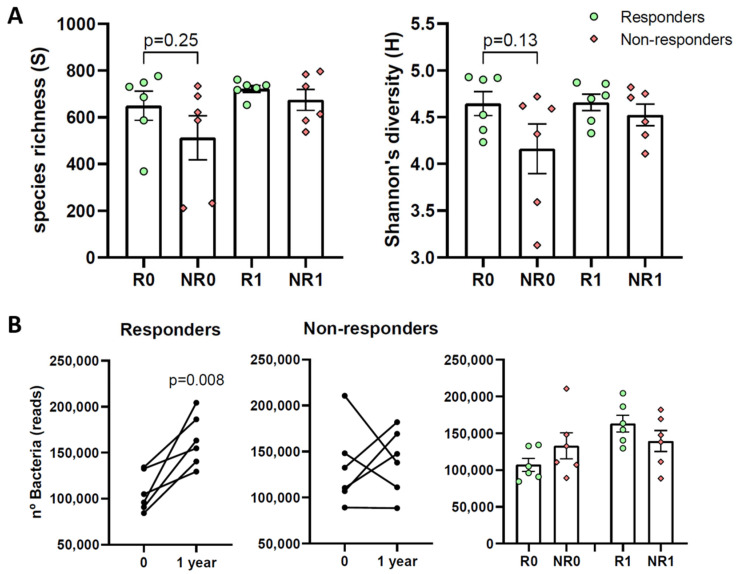
Microbiota diversity and number of bacteria in responder and non-responder BS patients. (**A**) Alpha diversity: number of species (S, left) and Shannon index (H, right). (**B**) Total number of bacteria. Analyses were performed at the time of surgery (R0 and NR0) or one year later (R1 and NR1). Green dots, responders; red diamonds, non-responders. Paired ((**B**) left) or unpaired ((**A**) and (**B**) right) *t*-tests were applied to determine statistical significance.

**Figure 3 nutrients-15-03187-f003:**
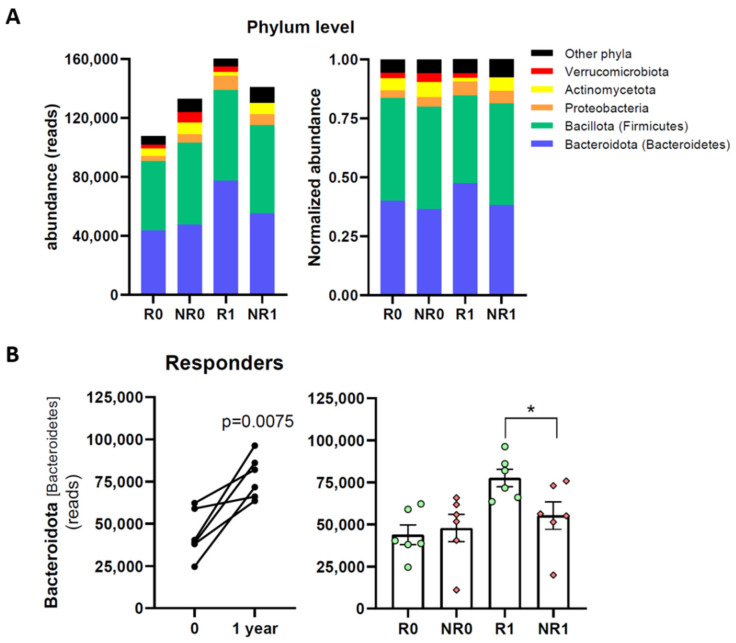
Fecal microbiota phyla in responder and non-responder BS patients. (**A**) Stacked bar chart of the more abundant phyla. (**B**) Abundance of phylum Bacteroidota. Analyses were performed at the time of surgery (R0 and NR0) or one year later (R1 and NR1). Green dots, responders; red diamonds, non-responders. Paired ((**B**) left) or unpaired ((**B**) right) *t* tests were applied to determine statistical significance; * *p* < 0.05.

**Figure 4 nutrients-15-03187-f004:**
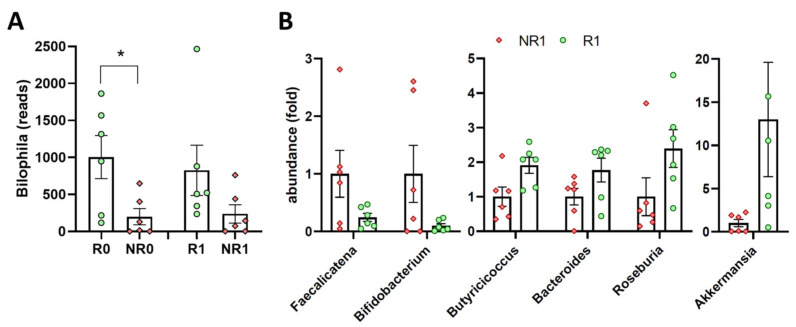
Fecal microbiota genera in responder and non-responder BS patients. (**A**) Analysis of the genus Bilophila. Unpaired *t*-test showed significant differences (* *p* < 0.05) at time 0. Analyses were performed at the time of surgery (R0 and NR0) or one year later (R1 and NR1). (**B**) Analysis of genera showing differences one year after BS (see statistical significance in Table 2). Green dots, responders; red diamonds, non-responders.

**Figure 5 nutrients-15-03187-f005:**
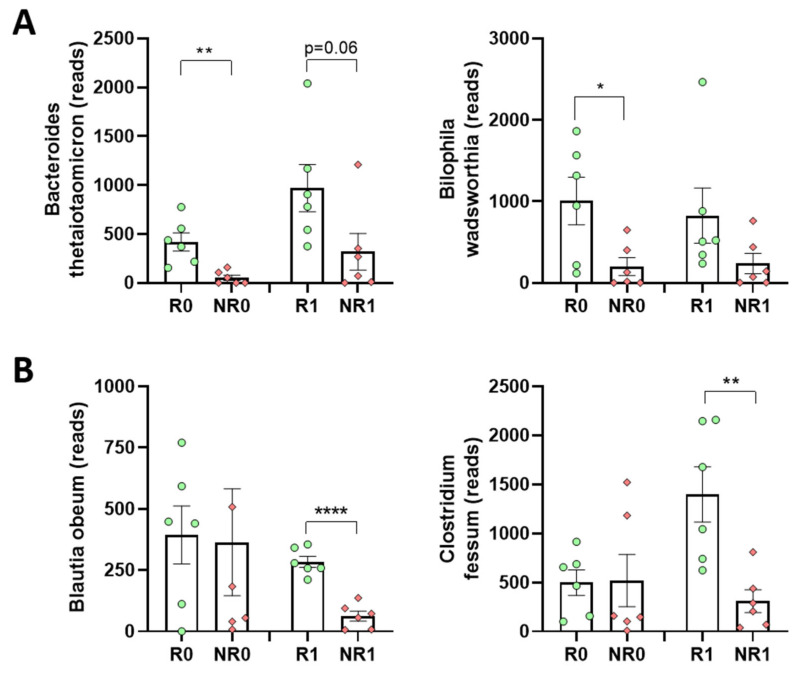
Fecal bacterial species in responder and non-responder BS patients. (**A**) Representative species showing significant differences at time 0. (**B**) Representative species showing significant differences one year after BS. Analyses were performed at the time of surgery (R0 and NR0) or one year later (R1 and NR1). Green dots, responders; red diamonds, non-responders. Statistical significance was determined by unpaired *t*-tests; * *p* < 0.05; ** *p* < 0.01; **** *p* < 0.0001.

**Figure 6 nutrients-15-03187-f006:**
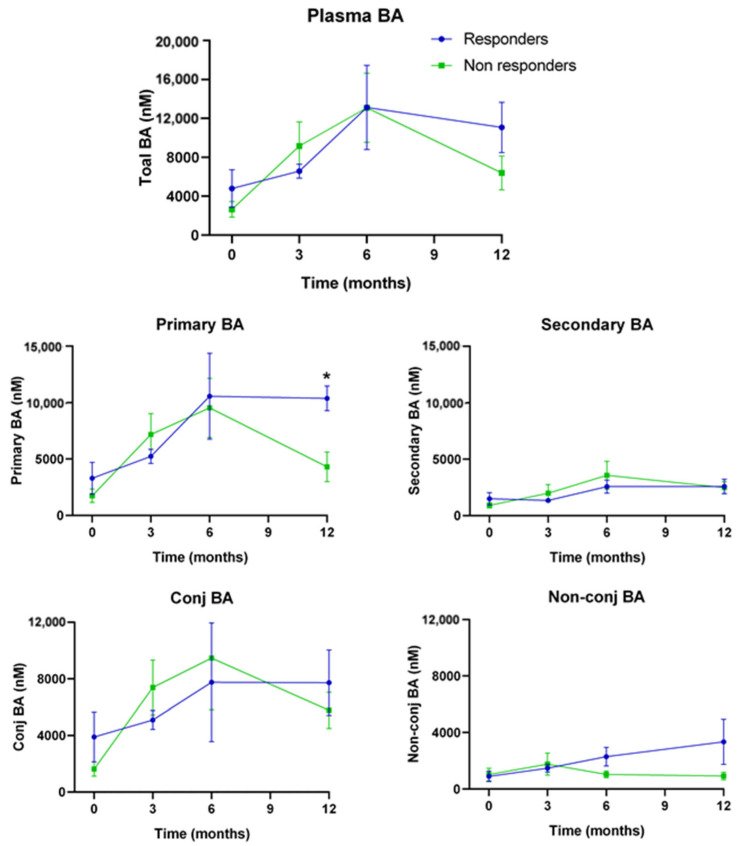
Time-course of plasma BA after BS in responder and non-responder patients. Concentration of BA was determined by UPLC-MS/MS in plasma samples at 0, 3, 6, and 12 months after surgery. Statistical significance was determined by unpaired *t*-tests; * *p* < 0.05.

**Figure 7 nutrients-15-03187-f007:**
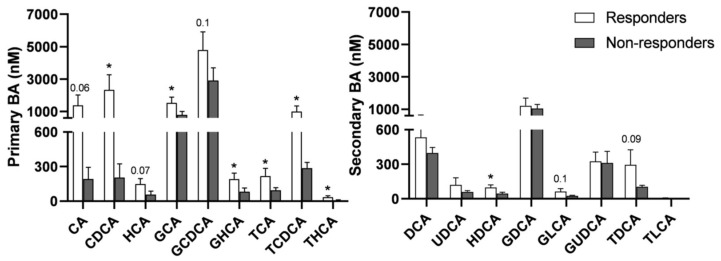
Concentration of individual plasma BA species in responders and non-responders. Concentration of BA was determined by UPLC-MS/MS in plasma samples at 12 months after surgery. Statistical significance was determined by unpaired *t*-tests; * *p* < 0.05. Cholic acid (CA), Chenodeoxycholic acid (CDCA), Hyocholic acid (HCA), Deoxycholic acid (DCA), Ursodeoxycholic acid (UDCA), Hyodeoxycholic acid (HDCA), Lithocholic acid (LCA). G, glycine conjugated BA; T, taurine conjugated BA.

**Table 2 nutrients-15-03187-t002:** *p* values of differences observed at different taxonomic levels.

	R1-R0	NR1-NR0	R0-NR0	R1-NR1
**PHYLUM**				
Bacteroidota	0.007 (↑R1)	0.346	0.698	0.044 (↑R1)
Proteobacteria	0.023 (↑R1)	0.631	0.465	0.332
Actinomycetota	0.054 (↓R1)	0.948	0.466	0.144 (↓R1)
Verrucomicrobiota	0.778	0.330	0.509	0.015 (↑R1)
**CLASS**				
Bacteroidia	0.008 (↑R1)	0.442	0.822	0.035 (↑R1)
Bacilli	0.051 (↑R1)	0.101	0.261	0.373
Actinomycetes	0.075 (↓R1)	0.724	0.672	0.100 (↓R1)
Negativicutes	0.521	0.071 (↓NR1)	0.785	0.318
Verrucomicrobiae	0.776	0.330	0.509	0.015 (↑R1)
**ORDER**				
Bacteroidales	0.008 (↑R1)	0.443	0.821	0.036 (↑R1)
Lactobacillales	0.051 (↑R1)	0.291	0.260	0.600
Bifidobacteriales	0.075 (↓R1)	0.724	0.672	0.100 (↓R1)
Veillonellales	0.521	0.071 (↓NR1)	0.785	0.318
Verrucomicrobiales	0.776	0.330	0.509	0.015 (↑R1)
**FAMILY**				
Bacteroidaceae	0.025 (↑R1)	0.770	0.417	0.089 (↑R1)
Prevotellaceae	0.071 (↑R1)	0.986	0.559	0.655
Streptococcaceae	0.053 (↑R1)	0.113	0.227	0.701
Akkermansiaceae	0.776	0.330	0.509	0.015 (↑R1)
Bifidobacteriaceae	0.075 (↓R1)	0.724	0.672	0.100 (↓R1)
Veillonellaceae	0.521	0.071 (↓NR1)	0.785	0.318
Oscillospiraceae	0.691	0.066 (↓NR1)	0.932	0.211
Odoribacteraceae	0.030 (↑R1)	0.047 (↑NR1)	0.626	0.784
**GENUS**				
Bacteroides	0.025 (↑R1)	0.764	0.417	0.097 (↑R1)
Roseburia	0.054 (↑R1)	0.280	0.224	0.103 (↑R1)
Anaerobutyricum	0.015 (↑R1)	0.126	0.510	0.777
Prevotella	0.066 (↑R1)	0.967	0.439	0.668
Streptococcus	0.055 (↑R1)	0.114	0.227	0.700
Akkermansia	0.776	0.330	0.509	0.015 (↑R1)
Bifidobacterium	0.075 (↓R1)	0.724	0.672	0.100 (↓R1)
Oscillibacter	0.044 (↓R1)	0.207	0.247	0.234
Faecalicatena	0.389	0.696	0.774	0.100 (↓R1)
Butyricicoccus	0.402	0.014 (↓NR1)	0.557	0.030 (↑R1)
Butyricimonas	0.161	0.053 (↑NR1)	0.431	0.910
Parabacteroides	0.443	0.074 (↑NR1)	0.094 (↑R0)	0.300
Bilophila	0.698	0.812	0.027 (↑R0)	0.135

1 vs. 0: paired *t*-tests. R vs. NR: unpaired *t* tests. ↑ and blue numbers: increased abundance, ↓ and red numbers: decreased abundance.

**Table 3 nutrients-15-03187-t003:** Differences between responders and non-responders in the microbiome at the species level.

R0 vs. NR0	FC	*p* Values	R1 vs. NR1	FC	*p* Values
*Bacteroides bouchesdurhonensis*	4.2	0.063	*Bacteroides faecis*	2.9	0.099
*Bacteroides caccae*	4.6	0.020	*Bacteroides salyersiae*	24.8	0.077
*Bacteroides caecimuris*	12.4	0.030	*Bacteroides thetaiotaomicron*	3.0	0.059
*Bacteroides faecis*	9.8	0.047	*Bacteroides zhangwenhongii*	5.9	0.055
*Bacteroides finegoldii*	3.6	0.014	*Blautia brookingsii*	5.2	0.0003
*Bacteroides koreensis*	5.3	0.055	*Blautia faecicola*	1.8	0.065
*Bacteroides thetaiotaomicron*	7.8	0.0031	*Blautia obeum*	4.6	0.00003
*Bacteroides xylanisolvens*	15.1	0.084	*Butyricicoccus faecihominis*	2.0	0.030
*Bacteroides zhangwenhongii*	9.3	0.021	*Clostridium fessum*	4.5	0.005
*Bilophila wadsworthia*	5.0	0.027	*Clostridium pacaense*	2.1	0.061
*Blautia intestinalis*	4.7	0.063	*Faecalibacterium longum*	2.9	0.072
*Clostridium geopurificans*	4.1	0.051	*Fusobacterium animalis*	3.4	0.025
*Olsenella intestinalis*	4.3	0.090	*Roseburia hominis*	5.2	0.022
*Oscillibacter valericigenes*	4.1	0.068			
*Parabacteroides distasonis*	5.7	0.063			

FC: Fold-change.

## Data Availability

The data presented in this study are available on request from the corresponding author. The data are not yet publicly available as this is a pilot study of a larger ongoing project. Data will be deposited in an open data repository when the project is finished.

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
