# Peer review of "Gut Microbiota and Plasma Bile Acids Associated with Non-Alcoholic Fatty Liver Disease Resolution in Bariatric Surgery Patients"

_nutrients, 2023, doi:10.3390/nu15143187_

Round 1
Reviewer 1 Report
The manuscript by Pérez-Rubio and coworkers reports on differences on gut microbiota and bile acid composition in patients that underwent bariatric surgery (BS), claiming that these differences contribute to the response to the surgery in terms of non-alcoholic fatty liver disease (NAFLD).
Although the idea is very interesting and the authors found some evidence that support their rationale, the study is very much limited by the small number of patients that were investigated. This is a big problem and it reduces the credibility of the conclusions.
Below there is a list of points that are hampering the interpretation of the results:
1. Table 1. The subjects in Responders (R) and Non-responders (NR) groups are paired in terms of age, BW and BMI. The average value of LDL-cholesterol is 25% lower in R compared to NR, while triglycerides are 30% higher in R, compared to NR. Insulin average values are 41% lower in R compared to NR. The statistical test does not pick up these differences, probably due to the low number of patients in each group, but these differences are too large to be ignored.
2. Figure 1. There is a very large difference in the % of liver fat in both groups. In pane A of Fig 1, the authors even used a different scale on Y axys of the graphs, which can be misleading. It is very clear that NR have much less fat in the liver, when compared to R – at the time of surgery, the average liver fat % in R seem to be around 10%, while it is ~5% in the NR. It has been demonstrated in several studies that patients with a more dysregulated metabolism are those who profit from interventions. So, it comes as no surprise that subjects with less liver fat are those that do not respond to the BS. NR also had a smaller change in BMI, in comparison to R, which might also help explain the poor outcome in terms of change in % liver fat. The authors do not explore this possibility in the text.
3. Figure 2. It is also clear, looking at the data points in pane A that there are 2 patients that have low species richness and Shannon’s index. If these 2 individuals are not considered, there are no differences between R and NR in these variables.
4. Figure 3. I do agree with the authors that there is an increase in the reads of Bacteroidota in R after 1 year, compared to the time of surgery (Pane A). But there is also an increase in the reads of Proteobacteria, which the authors do not mention in the discussion. In this case, the difference seems to be even larger. In Pane B, it looks like there is only one patient with very low reads for Bacteroidota in NR0 and NR1. If that individual is taken out, it is likely that there will be no difference between the groups.
5. Figure 4. The heatmap showing average reads maks interindividual differences. I would like to see the heatmap showing the results for each individual.
6. Figure 6. The increase in plasma bile acids (BA) is evident and it is a nice observation. It is not clear whether the plasma samples were collected before surgery or in the days after. This information is very important for the interpretation of these results, as the dietary habits of the patients can play a big role in these results. The same is also true for the microbiota results. Apart from that, BA are known for their large variability amongst individuals. The increase in non-conjugated BA at month 12 in responders needs to be investigated more carefully. The authors should present individual values, rather than the averages. Perhaps looking into the sum of unconjugated, primary BA would allow capturing the difference between the 2 groups.
Author Response
Response to Reviewer 1
Comments and Suggestions for Authors
The manuscript by Pérez-Rubio and coworkers reports on differences on gut microbiota and bile acid composition in patients that underwent bariatric surgery (BS), claiming that these differences contribute to the response to the surgery in terms of non-alcoholic fatty liver disease (NAFLD).
Although the idea is very interesting and the authors found some evidence that support their rationale, the study is very much limited by the small number of patients that were investigated. This is a big problem, and it reduces the credibility of the conclusions.
Below there is a list of points that are hampering the interpretation of the results:
- Table 1. The subjects in Responders (R) and Non-responders (NR) groups are paired in terms of age, BW and BMI. The average value of LDL-cholesterol is 25% lower in R compared to NR, while triglycerides are 30% higher in R, compared to NR. Insulin average values are 41% lower in R compared to NR. The statistical test does not pick up these differences, probably due to the low number of patients in each group, but these differences are too large to be ignored.
We agree that the number of patients is small. We acknowledged this in lines 71-72 “..we investigated the fecal microbiota and plasma BA profiles in two small groups of BS patients..”. Moreover, in the revised version, we emphasize this fact by describing this study as a pilot study of a larger ongoing project (lines 80 and 149).
Working with small sizes could lead to a lack of significance when differences are small. However, not only averages (means) are important, the variability (SD) is also determinant. Thus, for the case of LDL-cholesterol (86.4 in R0 vs 114.7 in NR0, and a pooled SD of 56) the estimated sample size to achieve significance would be 124 patients (62 in each group). Thus, the Reviewer is right that these differences should not be ignored, but the large number of patients required makes almost impossible to demonstrate their significance.
Following the Reviewer’s recommendation, we have commented on these differences in lines 152-154.
- Figure 1. There is a very large difference in the % of liver fat in both groups. In pane A of Fig 1, the authors even used a different scale on Y axys of the graphs, which can be misleading. It is very clear that NR have much less fat in the liver, when compared to R – at the time of surgery, the average liver fat % in R seem to be around 10%, while it is ~5% in the NR. It has been demonstrated in several studies that patients with a more dysregulated metabolism are those who profit from interventions. So, it comes as no surprise that subjects with less liver fat are those that do not respond to the BS. NR also had a smaller change in BMI, in comparison to R, which might also help explain the poor outcome in terms of change in % liver fat. The authors do not explore this possibility in the text.
The Reviewer is right, the average values of % of liver fat in R0 and NR0 groups are not the same. This was clearly shown in Table 1: 12.9% vs 7.7%. Nevertheless, we regret the mistake in the graph and have corrected the Y axis scales in Fig 1A to be equal. It can also be seen (in Table 1 and Fig 1A) that data in these two groups have a large variability so that points in both groups completely overlap. No statistically significant differences were found. In scientific terms, these two groups are not likely different.
It is also true that patients with a more deregulated metabolism can benefit more (in relative terms, i.e. fold-change). However, the important point here is that patients of the NR group did not respond to the intervention. Their % of liver fat remained at the same level, so that it was 4.4-fold higher (and statistically significant) in NR after one year (NR1 vs R1).
The opposite can be said for BMI. Both groups reduced BMI significantly, although, after one year, the reduction in R1 was somewhat greater than in NR1. However, points in both groups had extensive overlap and this small difference was not statistically significant.
- Figure 2. It is also clear, looking at the data points in pane A that there are 2 patients that have low species richness and Shannon’s index. If these 2 individuals are not considered, there are no differences between R and NR in these variables.
The Reviewer is right, and in agreement, we clearly stated that “no significant differences were observed” because p> 0.05 (lines 193 and 196). We only mention that there is a trend in NR0 to lower richness and Shannon index, and this trend is still observed if we remove the 2 lowest values in NR0.
- Figure 3. I do agree with the authors that there is an increase in the reads of Bacteroidota in R after 1 year, compared to the time of surgery (Pane A). But there is also an increase in the reads of Proteobacteria, which the authors do not mention in the discussion. In this case, the difference seems to be even larger. In Pane B, it looks like there is only one patient with very low reads for Bacteroidota in NR0 and NR1. If that individual is taken out, it is likely that there will be no difference between the groups.
Thanks for your observation regarding Proteobacteria. You are right, this phylum had an increase of 3-fold form R0 to R1 (p=0.023). However, we are more interested in differences between responders and non-responders, and, when we looked at Proteobacteria the differences between R1 and NR1 were not statistically significant (see Table 2).
We also agree that in Figure 3 there is one patient with lower reads in NR. However, if this patient is left out the conclusions are very similar: the difference between R1 and NR1 goes from 29% to 20%, being R1 still higher than NR1.
- Figure 4. The heatmap showing average reads maks interindividual differences. I would like to see the heatmap showing the results for each individual.
A new Figure 4B, showing individual values, has been made (page 9). Statistical differences are always shown in Table 2.
- Figure 6. The increase in plasma bile acids (BA) is evident and it is a nice observation. It is not clear whether the plasma samples were collected before surgery or in the days after. This information is very important for the interpretation of these results, as the dietary habits of the patients can play a big role in these results. The same is also true for the microbiota results. Apart from that, BA are known for their large variability amongst individuals. The increase in non-conjugated BA at month 12 in responders needs to be investigated more carefully. The authors should present individual values, rather than the averages. Perhaps looking into the sum of unconjugated, primary BA would allow capturing the difference between the 2 groups.
Thanks for the comment. In the revised version, we indicate that time 0 plasma was collected on the day of intervention, just before surgery (line 345). All the other plasma samples were collected at different times after surgery (3, 6 or 12 months). The same applies for fecal sample collection.
Dietary indications for these patients are very precise and standardized both before and after surgery. Nevertheless, as the Review comments, BA variability is large, something that can be appreciated by looking at the error bars in Figures 6 & 7.
We attempted to represent individual values in the BA figures, but our software version of GraphPad Prism does not allow to represent lines, means, errors, and individual data points in the same plot. The only option is to represent individual data points and lines, and the result was quite unclear. Mixed clouds were formed in many of the time points, which made it difficult to follow the time-dependency of R vs NR. So, for the sake of clarity, we prefer to keep the representation of the mean ± SEM.
We would like to thank the Reviewer for the advice of looking into the sum of primary unconjugated BA. This was a good idea as the difference between R and NR is larger (8.2-fold) and with a p value of 0.1. We have incorporated this novel info in lines 353-355.

Reviewer 2 Report
The authors investigated the fecal microbiota and plasma bile acid profiles in two small groups of bariatric surgery patients (responders and non-responders), at the time of surgery and 1 year later. The topic is interesting and the manuscript is well-writen. However, the reviewer has some comments/concerns and recommended major revisions.
Major concerns:
1. Was the clinical trial registered in an ICMJE-approved public trials registry at or before the onset of participant enrolment?
2. The authors mainly used paired t test for pre-post comparison within group and unpaired t test for post comparison between the groups. However, the current t test methods didn’t adjust for pre-test measurements and difference between the two groups. It is highly recommended to use repeated measures ANOVA or ANCOVA for two groups pre-post comparisons. Alternatively, a between group t test of the pre-post change values may also work.
3. How was the sample size estimated? The sample size may not be enough to test so many endpoints. It may be a weakness of this paper.
4. Current statistical analysis may not be proper for metagenomic data. The authors could consider LEfSe or other bioinformatic tool for metagenomic data analysis.
Specific points:
Figure 1: Do the subject have same ethnicity or not? If not, it is recommended to present subjects’ ethnicity in Figure 1.
Line 111: Since FastQC was performed, what is the quality threshold for data analysis? Did you filter out the low-quality data?
Line 121: It is recommended to perform beta-diversity analysis to see how subjects clustered before and after treatment.
n/a
Author Response
Comments and Suggestions for Authors
The authors investigated the fecal microbiota and plasma bile acid profiles in two small groups of bariatric surgery patients (responders and non-responders), at the time of surgery and 1 year later. The topic is interesting and the manuscript is well-writen. However, the reviewer has some comments/concerns and recommended major revisions.
Major concerns:
- Was the clinical trial registered in an ICMJE-approved public trials registry at or before the onset of participant enrolment?
Thanks for this comment. We likely did not explain properly the type of clinical study. Our study is not a clinical trial as defined by ICMJE: any research study that prospectively assigns patients/participants to one or more health-related interventions to evaluate the effects on health outcomes. Our study is an observational study, in which all patients are morbidly obese patients subjected to bariatric surgery. We do not have different health-related interventions, only retrospectively (one year after surgery) we divided patients into responders and non-responders based on their % liver fat. This type of observational studies does not require ICMJE registration (https://www.icmje.org/about-icmje/faqs/clinical-trials-registration/). We have added more information regarding our study in point 2.1 lines 94-96.
- The authors mainly used paired t test for pre-post comparison within group and unpaired t test for post comparison between the groups. However, the current t test methods didn’t adjust for pre-test measurements and difference between the two groups. It is highly recommended to use repeated measures ANOVA or ANCOVA for two groups pre-post comparisons. Alternatively, a between group t test of the pre-post change values may also work.
Thanks to the Reviewer for this suggestion. If I’m not wrong a repeated measures ANOVA is used to compare means of three or more groups. Usually, this would occur when a participant is repeatedly tested more than two times. In most of our analyses we have only measured two times: 0 (surgery) and one year. In this case, one would typically use a mixed 2 (between) x 2 (within) ANOVA. In other words, a 2-way ANOVA in which one factor is group and the other factor is time. This statistical approach however moves away the main focus of our study: to find differences between responders and non-responders. For us, the pre-post change within each group is not the main focus. It is only interesting if it contributes to explain differences between responders and non-responders one year after surgery. Indeed, most of our conclusions would have been the same if we had only collected information one year after surgery (two groups R1 and NR1).
Nevertheless, we have tested several of our comparisons by using two-way ANOVA and found that results are very similar. For instance, according with unpaired t-test the difference of the Bacteroidota phylum between R1 and NR1 is significant (p=0.044, Table 2). If we apply two-way ANOVA and the two-stage linear step-up procedure of Benjamini, Krieger and Yekutieli for multiple comparisons, the difference between R1 and NR1 is also significant (p=0.035). In the case of the genus Bilophila results by unpaired t-test and two-way ANOVA are almost identical (0.0274 vs 0.0271).
Therefore, we would prefer continuing with the simpler statistical approach, which allows us to stress the importance of the differences between responders and non-responders, the main objective of this study.
- How was the sample size estimated? The sample size may not be enough to test so many endpoints. It may be a weakness of this paper.
Our sample size is small, as this study is a pilot study of a larger ongoing project. We have emphasized this fact more clearly in lines 80 and 149. Nevertheless, sample size was estimated with the idea of comparing two independent means (NR vs R):
- Regarding liver steatosis (an important variable in this study), we anticipated a difference in MRI-PDFF of around 5% and a pooled SD of 3%. In these conditions the calculate samples size is 6 for each group.
- Regarding metagenomic data we expected differences between NR and R of 3500-350 reads (depending on the taxonomic level) and a pooled SD of 2000-200 reads, which also gave us a sample size 6 for each group.
Calculation of sample size was based on https://statulator.com/SampleSize/ss2M.html#
- Current statistical analysis may not be proper for metagenomic data. The authors could consider LEfSe or other bioinformatic tool for metagenomic data analysis.
We are aware that the statistical analyses performed may not seem the most appropriate. It is important to remind that our sample size is small and some metagenomic analyses also include a small number of features (e.g. only 8 phyla and 15 classes). This is not big data and, therefore, separate univariate analyses may be acceptable.
We would like to thank the Reviewer for the suggestion of using LEfSe. We found LEfSe to be an excellent tool for metagenomic data, very convenient and with multiple advantages. Nevertheless, if we compare LEfSe results (for example at the species level, NR1 vs R1) with our previous results (for example Table 3, NR1 vs R1) we found total concordance, so that all species selected by LEfSe are also included in our Table 3:
We have also performed multivariate PLSDA analyses to find discriminant features (NR1 vs R1) at the species level, and results again demonstrated total concordance with our results in Table 3
Thus, it is possible that, for small datasets, multivariate analysis approaches have similar performance than separate univariate analyses.
Specific points:
Figure 1: Do the subject have same ethnicity or not? If not, it is recommended to present subjects’ ethnicity in Figure 1.
Ten patients were Caucasian and two were Amerindian. This information has been included in the revised version (lines 82-83). In Figure 1 (upper panels), ethnicity is represented with dots of different colors.
Line 111: Since FastQC was performed, what is the quality threshold for data analysis? Did you filter out the low-quality data?
Yes, quality filtering on the raw tags was performed with FastQC with a threshold of 20 (Phred Score). Sequences below this threshold and chimeric reads were eliminated. Sequences with low quality ends (below the same threshold) were also trimmed, and general denoising was applied. No samples with any adapter contamination > 0.1% were found.
Line 121: It is recommended to perform beta-diversity analysis to see how subjects clustered before and after treatment.
We agree with the Reviewer, and it is something we should have done when metagenomic analyses were performed. Unfortunately, the bioinformatics expert that did these analyses is no longer available. If we want to have this analysis done, we need to pay a new whole analysis from raw data by a new bioinformatics service. They have a long waiting list, and the revision of this manuscript would be delayed well beyond the deadline given to us by the journal editors. Sorry for that.

Reviewer 3 Report
In the current study, authors explored the possibility of beneficial role of fecal microbiota in NAFLD resolution in morbidly obese patients undergoing BS. Although, it is an interesting study there are number of issues that need to be addressed before it can be accepted for publication.
· Authors need to check the manuscript for typos and English.
· Some details regarding the rationale of experiments should be provided in the beginning of each result section. It will help the readers in understanding. “It has been proven that the greater the diversity of the microbiota the 182 better the health outcomes.” Is very vague.
· Type of error bar (standard deviation or standard error of mean) should be mentioned in the figure legends. Figure 6 should also include individual data points.
· Authors should include an outlook at the end.
Authors need to check for typos and improve English. Scientific writing should also be improved.
Author Response
Response to Reviewer 3
Comments and Suggestions for Authors
In the current study, authors explored the possibility of beneficial role of fecal microbiota in NAFLD resolution in morbidly obese patients undergoing BS. Although, it is an interesting study there are number of issues that need to be addressed before it can be accepted for publication.
Authors need to check the manuscript for typos and English.
Thanks for your comment. Typos have been checked, and we also made an effort to improve scientific English.
Some details regarding the rationale of experiments should be provided in the beginning of each result section. It will help the readers in understanding. “It has been proven that the greater the diversity of the microbiota the better the health outcomes.” Is very vague.
The Reviewer is right, thanks for the suggestion. We revised the Result section to include short explanatory sentences on the rationale. Moreover, we have rewritten the mentioned sentence to be clearer (lines 189-190).
Type of error bar (standard deviation or standard error of mean) should be mentioned in the figure legends. Figure 6 should also include individual data points.
In “2.4. Statistics” we have explained that in all the figures the error bars represent the standard error of mean, whereas only in Table 1 the error is shown as standard deviation.
Regarding Figure 6, our software version of GraphPad Prism does not allow to represent lines, means, errors, and individual data points in the same plot. The only option is to represent individual data points with lines, and the result was quite unclear. Mixed clouds were formed in many of the time points, making it difficult to follow the time-dependency of R vs NR. So, for the sake of clarity, we prefer to keep the representation of the mean ± SEM.
Authors should include an outlook at the end.
A brief sentence on the perspective of this study has been included at the end of the discussion (lines 516-520).

Reviewer 4 Report
This paper compares the composition of the gut microbiota and plasma bile acids in 12 patients at the time of bariatric surgery and 12 months later. Patients were divided into two groups resolving (“responders”) or not (non-responders”) their non-alcoholic fatty liver disease.
Results show that responders have a greater abundance of some bacterial species and a decrease in others, as compared to non-responders. They also indicate an increase in plasma primary bile acids BA, which could result from a reduction in bacterial gut species capable of generating secondary BA. The authors propose that this could explain the effect on NAFLD.
This work appears well performed and clearly presented and discussed. It also provides interesting data to further analyze the benefits of bariatric surgery and, therefore, deserves to be published.
Nevertheless, I have two important questions and some minor criticisms that should be answered by the authors.
Question 1: as a non-biostatistician, I wonder if such a small number of patients permits the authors to state: “our results on BA allow us to conclude that NAFLD resolution…”. I would prefer: “our results on BA suggest that NAFLD resolution”. The authors should comment on that.
Question 2: they wrote: “Signalling of primary BA, such as CDCA, through nuclear receptors, such as FXR, could contribute to a more efficient NAFLD resolution.” This should be discussed more deeply and experimental approach to test it should be proposed.
Minor criticisms.
· NAFLD for non-alcoholic fatty liver disease must be defined in the abstract (and in the title?).
· CDCA for chenodeoxycholic acid must be defined on its first appearance in the text.
· L342: “However, excessive SCFA may also have harmful effects [9]”. This should be further explained.
· L 392: “Bile acids stand out among the bacterial products that have a significant effect on host metabolism.” Since bile acids are abundantly produced by the liver and metabolized in the gut, this statement should be explained or modified.
· Fig. 1A: to improve readability, please use the same ordinate scale in “responders” and “non-responders”.
· “The knowledge gained may help in designing interventions with prebiotics to guarantee the maximal rates of NAFLD resolution”, the last sentence of the abstract” should be developed in the discussion.
-
Author Response
Response to Reviewer 4
Comments and Suggestions for Authors
This paper compares the composition of the gut microbiota and plasma bile acids in 12 patients at the time of bariatric surgery and 12 months later. Patients were divided into two groups resolving (“responders”) or not (non-responders”) their non-alcoholic fatty liver disease.
Results show that responders have a greater abundance of some bacterial species and a decrease in others, as compared to non-responders. They also indicate an increase in plasma primary bile acids BA, which could result from a reduction in bacterial gut species capable of generating secondary BA. The authors propose that this could explain the effect on NAFLD.
This work appears well performed and clearly presented and discussed. It also provides interesting data to further analyze the benefits of bariatric surgery and, therefore, deserves to be published.
Nevertheless, I have two important questions and some minor criticisms that should be answered by the authors.
Question 1: as a non-biostatistician, I wonder if such a small number of patients permits the authors to state: “our results on BA allow us to conclude that NAFLD resolution…”. I would prefer: “our results on BA suggest that NAFLD resolution”. The authors should comment on that.
Thanks for the comment. The reviewer is right. Our sample size is small. We have commented on this in lines 72 and 149. We have also added that “this study is a pilot observational study of a larger ongoing project” (line 80). Therefore, we agree that we should be cautious with the conclusions. We have corrected the sentence according with the Reviewer’s suggestion (line 502).
Question 2: they wrote: “Signalling of primary BA, such as CDCA, through nuclear receptors, such as FXR, could contribute to a more efficient NAFLD resolution.” This should be discussed more deeply and experimental approach to test it should be proposed.
Thanks for the recommendation. It is a very interesting topic and agree that it should be properly discussed. See it in lines 505-513.
Minor criticisms.
NAFLD for non-alcoholic fatty liver disease must be defined in the abstract (and in the title?).
Done.
CDCA for chenodeoxycholic acid must be defined on its first appearance in the text.
Done. We have included the definition of all BA abbreviations (lines 363-364 and Figure 7 legend).
L342: “However, excessive SCFA may also have harmful effects [9]”. This should be further explained.
SCFA increase insulin sensitivity and lipid oxidation, and reduce hepatic fat storage, chylomicron secretion and liver inflammation. Moreover, SCFAs help to maintain the gut barrier permeability and decrease LPS levels. However, excessive SCFAs may inhibit AMPK in the liver and increase the accumulation of hepatic free fatty acids, while inducing pro-inflammatory T cells.
We have explained the sentence in lines 395-397.
L 392: “Bile acids stand out among the bacterial products that have a significant effect on host metabolism.” Since bile acids are abundantly produced by the liver and metabolized in the gut, this statement should be explained or modified.
We have changed this statement with a longer more explanatory sentence (lines 447-452).
Fig. 1A: to improve readability, please use the same ordinate scale in “responders” and “non-responders”.
Done
“The knowledge gained may help in designing interventions with prebiotics to guarantee the maximal rates of NAFLD resolution”, the last sentence of the abstract” should be developed in the discussion.
Done. See the last sentence of the discussion in lines 516-520.

Round 2
Reviewer 1 Report
The authors incorporated significant changes in the manuscript, better describing some points and improving data presentation.
The authors acknowledge that the sample size is small and that the study is a pilot. This is important as it serves to start the discussion, without making a too strong point.
Reviewer 2 Report
The authors addressed most of my concerns and the manuscript is acceptable for publication.
n/a
Reviewer 3 Report
Revised manuscript is significantly improved and can be accepted for publication.
Reviewer 4 Report
The manuscript has been significantly improved and answers have been given to the referee requests.
No further objection.